# Does G Protein-Coupled Estrogen Receptor 1 Contribute to Cisplatin-Induced Acute Kidney Injury in Male Mice?

**DOI:** 10.3390/ijms23158284

**Published:** 2022-07-27

**Authors:** Eman Y. Gohar, Rawan N. Almutlaq, Chunlan Fan, Rohan S. Balkawade, Maryam K. Butt, Lisa M. Curtis

**Affiliations:** 1Division of Nephrology and Hypertension, Vanderbilt University Medical Center, Nashville, TN 37232, USA; 2Division of Nephrology, Department of Medicine, University of Alabama at Birmingham, Birmingham, AL 35294, USA; almut051@umn.edu (R.N.A.); chunlanfan@uabmc.edu (C.F.); rohanbalkawade@uabmc.edu (R.S.B.); maryam27@uab.edu (M.K.B.); lisacurtis@uabmc.edu (L.M.C.)

**Keywords:** GPER30, KIM-1, heme oxygenase-1, inflammation, apoptosis

## Abstract

Nephrotoxicity is the dose-limiting side-effect of the chemotherapeutic agent cisplatin (Cp). Recent evidence points to renal protective actions of G protein-coupled estrogen receptor 1 (GPER1). In addition, it has been shown that GPER1 signaling elicits protective actions against acute ischemic injuries that involve multiple organ systems; however, the involvement of GPER1 signaling in Cp-induced acute kidney injury (AKI) remains unclear. This study tested whether genetic deletion of GPER1 exacerbates Cp-induced AKI in male mice. We subjected male mice, homozygous (homo) and heterozygous (het) knockout for the GPER1 gene, and wild-type (WT) littermates to Cp or saline injections and assessed markers for renal injury on the third day after injections. We also determined serum levels of proinflammatory markers in saline and Cp-treated mice. Given the protective role of heme oxygenase-1 (HO-1) in Cp-mediated apoptosis, we also investigated genotypic differences in renal HO-1 abundance, cell death, and proliferation by Western blotting, the TUNEL assay, and Ki67 immunostaining, respectively. Cp increased serum creatinine, urea, and neutrophil gelatinase-associated lipocalin (NGAL) levels, the renal abundance of kidney injury molecule-1, and NGAL in all groups. Cp-induced AKI resulted in comparable histological evidence of injury in all genotypes. WT and homo mice showed greater renal HO-1 abundance in response to Cp. Renal HO-1 abundance was lower in Cp-treated homo, compared to Cp-treated WT mice. Of note, GPER1 deletion elicited a remarkable increase in renal apoptosis; however, no genotypic differences in cell proliferation were observed. Cp augmented kidney Ki67-positive counts, regardless of the genotype. Overall, our data do not support a role for GPER1 in mediating Cp-induced renal injury. GPER1 deletion promotes renal apoptosis and diminishes HO-1 induction in response to Cp, suggesting that GPER1 may play cytoprotective and anti-apoptotic actions in AKI. GPER1-induced regulation of HO-1 and apoptosis may offer novel therapeutic targets for the treatment of AKI.

## 1. Introduction

Cisplatin (Cp) is an effective chemotherapeutic agent used to treat a wide variety of malignancies [1]. Cp is a platinum compound that is eliminated primarily via renal clearance. Nephrotoxicity is the dose-limiting side effect of Cp [2,3]. Acute kidney injury (AKI) following Cp administration is multi-factorial, involving cellular toxicity, inflammation, renal vasoconstriction, and proximal tubular injury [4,5,6]. Growing evidence implicates a role for sex steroids in the incidence, progression, and outcomes of AKI [7,8,9,10]. However, the contribution of estrogenic signaling in the mechanisms underlying Cp-induced AKI is unclear.

Estrogens exert their effects via activation of the classical estrogen receptors (ERα and ERβ) or the G protein-coupled estrogen receptor 1 (GPER1). GPER1 is a recently identified ER that can promote both rapid nongenomic signaling events as well as transcriptional regulation [11,12,13]. GPER1 is ubiquitously expressed throughout the body including the kidneys [14]. Of note, within the kidney, GPER1 is expressed in the renal tubules as well as in renal epithelial cells [15,16]. Recent evidence points to a renal protective potential of GPER1 in nephropathies evoked by salt-induced hypertension [16,17] and ischemia reperfusion [18] in female rats. However, the involvement of GPER1 signaling in Cp-induced nephrotoxicity/AKI has not been explored.

Estrogens have been historically considered as solely female sex hormones. However, studies during the last two decades established a role for estrogens and their receptors in regulating reproductive [19,20] as well as nonreproductive biological functions in males. Emerging evidence indicates that GPER1 regulates a number of physiological processes in males. GPER1 has been shown to regulate spermatogenesis [21], vascular responsiveness [22,23,24], and cardiac function [25,26]. In addition, GPER1 exerts protective actions against pulmonary hypertension [27], obesity, and diabetes in male mice [28,29]. However, little information is available regarding the role of GPER1 in preserving renal function in males under physiological or pathophysiological conditions.

Although various recent studies have suggested that GPER1 signaling elicits protective actions against cardiac [25,30,31,32,33], cerebral [34,35], renal [18], hepatic [36], and intestinal [37] injury induced by ischemia reperfusion, whether GPER1 confers protection against Cp-induced kidney damage is not known. Therefore, this study was designed to test whether genetic deletion of GPER1 exacerbates AKI induced by Cp in male mice. Here, we subjected male mice, homozygous or heterozygous knockout for the GPER1 gene, and wild-type littermates to Cp injections and assessed markers for renal injury. Specifically, we measured serum creatinine (SCr), urea, neutrophil gelatinase-associated lipocalin (NGAL), histological indexes of renal tubular injury, and abundance of kidney injury molecule-1 (KIM-1) and NGAL in the kidney. We also measured the serum levels of proinflammatory markers in Cp and saline-treated mice of all the genotypes to identify the contribution of cytokines in Cp-induced AKI. Given the protective role of heme oxygenase-1 (HO-1) against cell death in Cp-mediated cell injury, we also investigated genotypic differences in HO-1 abundance, apoptosis, and cell proliferation in kidneys obtained from saline and Cp-treated mice.

## 2. Results

**Body weight loss and mortality rate.** GPER1 deletion decreased the body weight at baseline (Figure 1A). Mice, randomly assigned to the treatment, did not show basal body weight differences at the time of saline and Cp-treatment within any genotype (Figure 1A). Cp resulted in a gradual loss of body weight throughout the first 3 days after the injection (Figure 1B). Body weight loss and death after Cp injections were comparable in the three genotypes (Figure 1B,C, respectively). No significant differences in body weight or mortality occurred in the mice that received saline injections (Figure 1B,C).

**Renal injury.** All animals that received Cp injections elicted increases in SCr, serum urea, and serum neutrophil gelatinase-associated lipocalin (NGAL) levels, compared to corresponing saline-treated animals (Figure 2A–C). The increases in the serum level of these renal injury markers were similar in wild-type, het, and homo mice treated with Cp (Figure 2A–C).

Further analyses for renal cortical levels and histological indexes of renal tubular injury were performed. The abundance of renal cortical KIM-1, a biomarker for proximal tubule injury, was higher in saline-treated homo mice compared to the corresponding WT mice (14.00 ± 2.30 vs. 7.62 ± 0.70 AU, respectively; *p* = 0.0188; Student *t*-test; Figure 3A). No significant genotypic differences were shown in renal cortical NGAL level in saline-treated mice (Figure 3B). After Cp injections, renal cortical KIM-1 and NGAL protein levels were increased above corresponding saline-treated levels in all genotypes (Figure 3A–D). No differences in renal cortical KIM-1 or NGAL were observed between genotypes in Cp-treated mice (Figure 3A–D). PAS-staining of kidney sections revealed comparable structural morphological differences in Cp-treated mice (Figure 3E). Kidneys obtained from Cp-treated mice, regardless of animal genotype, elicited a loss of brush border and cast formation in the renal tubules (Figure 3E). Thus, kidney damage was evident on the third day following Cp injections, as demonstrated in the injury score quantitation (Figure 3F). Overall, these data suggest that GPER1 deletion did not change the extent of renal injury on the third day after Cp injections as assessed by serological, histological, and renal levels of kidney injury markers. 

**Serum levels of inflammatory cytokines.** Given the proinflammatory nature of Cp-induced AKI [4], we measured serum levels of cytokines in saline and Cp-treated WT, het, and homo mice. Cytokine panel assessment demonstrated an overall effect for Cp in increasing serum levels of tumor necrosis factor-α (TNF-α), interleukin-2 (IL-2), IL-5, and IL-10 (Figure 4A–C,E). Post hoc test analysis revealed that serum TNF-α levels were significantly higher in Cp-treated WT, het, and homo mice, compared to the corresponding saline-treated mice (Figure 4A), whereas serum IL-2 and IL-10 levels were significantly greater only in Cp-treated homo mice, compared to saline-treated mice of the same genotype (Figure 4B,E). Cp-induced upregulation of serum IL-2 and IL-10 did not reach statistical significance in WT or het mice using post hoc test analysis (Figure 4B,E). Importantly, GPER1 deletion lowered serum IL-12p70, regardless of whether the mice were treated with saline or Cp (Figure 4F). However, post hoc test analysis for IL-12p70 between WT and homo or het mice did not reach statistical significance for saline (*p* = 0.8946 and 0.1965, respectively) or Cp-treated mice (*p* = 0.8875 and 0.4900, respectively). No treatment or genotypic differences were observed in serum levels of IL-6, IL-1β, INF-γ (interferon-γ), and KC/GRO (Figure 4D,G–I). 

**HO-1 induction.** Western blotting revealed that kidney HO-1 levels (~28 kDa) were upregulated in response to Cp treatment in all genotypes (Figure 5A,B). The 28 kDa band for HO-1 has been shown to be modulated in Cp-induced nephrotoxicity [7]. Importantly, kidney HO-1 levels were significantly blunted in Cp-treated homo and het mice, compared to Cp-treated WT controls (Figure 5A,B). In contrast, no genotypic differences in kidney HO-1 levels (~28-kDa) were observed in saline-treated mice (Figure 5A,B). Thus, GPER1 deletion attenuated HO-1 upregulation evoked by Cp treatment. 

**Cell death and proliferation.** It has been previously shown that Cp evokes apoptosis of renal tubular cells [38,39]. To assess the role of GPER1 in the development of Cp-induced renal cell death, a TUNEL assay was performed for kidney sections obtained from Cp and saline-treated homo and WT mice. Treatment of WT mice with Cp induced a slight increase in apoptosis as demonstrated by the number of TUNEL-positive nuclei in the cortex and the outer medullary regions of Cp-treated WT mice, compared to saline-treated WT mice (11.6 ± 3.9 vs. 4.8 ± 2.2 TUNEL-positive nuclei/field, respectively). However, this Cp-induced increase in apoptosis did not reach statistical significance (*p* = 0.11, Student *t*-test). Interestingly, GPER1 deletion resulted in a striking increase in renal apoptosis as evident by the remarkably greater number of TUNEL-positive nuclei in both the renal cortex and outer medullary regions of kidneys obtained from saline and Cp-treated homo mice (568.8 ± 106.4 vs. 412.4 ± 157.7 TUNEL-positive nuclei/field, respectively). Kidneys from Cp-treated WT mice demonstrated a remarkably lower number of TUNEL-positive nuclei, compared to those from Cp-treated homo mice (Figure 6A,B). However, Cp treatment to homo mice did not evoke an additional increase in the number of TUNEL-positive nuclei (Figure 6A,B). Examination of kidney sections at a higher magnification revealed that most of the TUNEL-positive staining was specific to the renal tubular nuclei. 

Cell proliferation assessed by Ki67 immunostaining was elevated in Cp-treated WT and Cp-treated homo mice, relative to the corresponding saline-treated mice (Figure 6C,D). No genotypic differences were seen in Ki67-positive counts in saline or Cp-treated mice (Figure 6C,D). These data indicate that GPER KO mice exhibit a dramatic increase in cell death, whereas Cp treatment elicits increases in cell proliferation.

## 3. Discussion

Evidence points to a critical role for the novel estrogen receptor GPER1 in the maintenance of cardiovascular and kidney health [11,16,17,24,29,40]. To date, the contribution of GPER1 to Cp-induced AKI is unknown. The major finding of this study is that GPER1 deletion in male mice exacerbates renal apoptosis and diminishes the increase in renal HO-1 abundance in response to Cp, without impacting the extent of renal injury evoked by Cp. These data suggest that GPER1 elicits an antiapoptotic role and a permissive role for HO-1 cytoprotective actions in Cp-induced AKI in males. 

In the current study, treating male mice with Cp increased serum levels of creatinine, urea, and NGAL; renal levels of KIM-1 and NGAL; histological indexes of renal tubular injury, similar to previous observations documenting AKI in Cp-treated animals [6,7]. Nevertheless, GPER1 deletion did not exacerbate the Cp-induced kidney injury assessed by these injury markers. Thus, the present findings do not support a contribution for GPER1 in the extent of Cp-induced renal injury in male mice. These findings are consistent with those from studies by Hutchens and colleagues conducted in female mice [41], showing that estradiol (E_2_) treatment attenuates acute renal injury following cardiac arrest and cardiopulmonary resuscitation irrespective of GPER1 gene deletion. In contrast, it has been demonstrated that GPER1 activation by G1 reduces renal injury in female Dahl salt-sensitive rats [17], mRen2.Lewis rats [16], and Sprague Dawley rats [18]. Whether pharmacological interventions using the selective GPER1 agonists/antagonists, rather than genetic approaches, induce a renoprotective effect against Cp-induced AKI remains to be tested. It is unknown whether the global deletion of GPER1 may elicit compensatory changes in the other ER profile in the male kidney, which may account for the lack of renal protection against Cp-induced AKI. Furthermore, given that the expression of renal GPER is remarkably lower in males than females [14,40], it remains to be determined whether GPER1 elicits protective actions against Cp-induced AKI in females. 

Sex-specific patterns in AKI have been well documented [8]. Evidence points to the fact that E_2_ mediates the protective effect of female sex. Recent evidence demonstrates that the inhibition of estrogen sulfotransferase, a conjugating enzyme that deactivated estrogen, ameliorates ischemic AKI in male and female mice [42]. Treatment with E_2_ for 7 days reduces serum urea nitrogen and tubular cell death in female, but not male, mice following cardiac arrest and cardiopulmonary resuscitation [41]. A single dose of E_2_ administered prior to ischemia/reperfusion accelerates renal tubule regeneration in male rats [43,44]. In addition, it has been recently shown that a single dose of E_2_, but not chronic E_2_ treatment, protects against renal damage evoked by hemorrhagic shock [45]. In contrast, higher serum levels of E_2_ in humans are associated with a greater severity of concomitant AKI and predict the development of new AKI in septic shock patients [10]. It is possible that E_2_ may elicit paradoxical effects on renal injury outcomes based on whether it is administered in an acute versus chronic setting. Therefore, the specific timeline of E_2_ treatment during the pathogenesis of AKI deserves further investigation in a sex-specific manner.

A protective role of GPER1 has been demonstrated in different models of nephropathies that involve the proximal tubule. GPER1 protects against methotrexate-induced nephrotoxicity in renal proximal tubular epithelial cells [46]. Of note, Cheng et al. showed that GPER1 is expressed in the proximal convoluted tubule [15,16]. Indeed, GPER1 is colocalized with megalin in the brush border of the proximal tubule in female mRen2.Lewis rats [16]. GPER1 activation by G1 reduces proteinuria, albuminuria, and oxidative stress in mRen2.Lewis rats [16]. Similarly, we demonstrated that G1 attenuates the urinary excretion of protein, albumin, and KIM-1 in female Dahl salt-sensitive rats, via preserving the proximal tubule brush border integrity [17]. GPER1 translocates to the brush border membrane of the proximal tubule in male rats in response to aldosterone administration [47]. In female mice, GPER1 expression is localized to the basolateral surface during the proestrus phase of the estrus cycle and is redistributed intracellularly during the estrus phase, suggesting that the sex hormonal status regulates GPER1 translocation in the renal epithelia [15]. Together, complementary work is required to determine how sex and Cp treatment impact the subcellular distribution pattern and translocation of GPER1 in the renal epithelia.

In particular, GPER1 has been shown to mediate protective effects in different ischemia reperfusion models that involves multiple organ systems. Several studies revealed that GPER1 activation improves the functional recovery of the male and female rodent heart exposed to ischemia reperfusion [25,30,31,32,33]. Additionally, activation of GPER1 suppresses cerebral [34,35], hepatic [36], intestinal [37], and renal [18] injury following ischemia reperfusion. Moreover, GPER1 abundance increases in the renal interlobular artery after ischemia reperfusion of the kidney [18]. Inhibition of apoptotic activity [34,36] and inflammation [35], improvement of mitochondrial function [30,32,33], and regulation of nitric oxide synthase [18,37,44] have been proposed as mediators of GPER1-protective actions against ischemia reperfusion injury in different organ systems. Given the present findings, it is possible that the renal protective actions of GPER1 are more pronounced in ischemic rather than drug-induced nephrotoxicity.

HO-1 is a cytoprotective inducible enzyme that degrades toxic free heme and releases beneficial byproducts [48,49]. HO-1 is upregulated as a beneficial response in cells exposed to cytotoxic insults [50,51]. HO-1 induction confers protection against renal injury in different AKI models including ischemia/reperfusion [52,53], Cp [54,55], and lipopolysaccharides [56]. Knockout of HO-1 in proximal tubule cells promotes autophagy and apoptosis after Cp [54]. Restoration of proximal tubule HO-1 expression prevents the autophagic and apoptotic response to Cp [54]. The overexpression of HO-1 in mice attenuates Cp-induced apoptosis [55]. Further, HO-1 deficiency results in more severe renal injury in mice treated with Cp [55]. Distinctive sex-specific patterns in HO-1 induction in response to Cp treatment in mice have been demonstrated [7]; however, the contribution of GPER1 signaling to post-AKI HO-1 induction is unknown. 

The E_2_-mediated regulation of HO-1 activity and expression have been established in multiple experimental models. The cardiac HO-1 expression and activity are reduced in response to ovariectomy [57]. E_2_ increases HO-1 expression in aorta [58], prostatic stoma [59], endothelial progenitor cells [60], and macrophages [61], leading to decreased reactive oxygen species (ROS) levels. In addition, E_2_ inhibits the endothelin-induced inhibition of HO-1 expression in cardiomyoblasts [62]. Further, E_2_ upregulates lung HO-1 after hemorrhagic shock, which may contribute to the role in E_2_ in lung protection after trauma-hemorrhage [63]. Whether GPER1 contributes to E_2_ regulatory effects on HO-1 is not clear. The present study demonstrates for the first time that the lack of GPER1 attenuates the increase in renal HO-1 following Cp treatment in male mice. The current findings provide insights for the potential permissive role for GPER1 in mediating renal HO-1 induction in Cp-evoked AKI. However, the putative mechanism underlying GPER1-induced upregulation of renal HO-1 in Cp-induced AKI remains unknown.

It is well-established that the HO-1 in the renal tubules elicits cytoprotective and antioxidant properties in kidney injury [48,49,64]. Importantly, it has been shown that GPER1 plays a critical role in regulating reactive oxygen species production. GPER1 deletion promotes oxidative damage in cardiomyocytes and increases myocardial oxidative stress-related genes [65]. GPER1 activation ameliorates renal and cardiac injury induced by oxidative stress following methotrexate [46] and doxorubicin [66] treatment, respectively. In addition, GPER1 protects pancreatic β cells against oxidative injury and promotes islet cell survival [67]. Given the role of oxidative stress in AKI pathology, understanding the mechanisms by which GPER1 could impact HO-1 and oxidative stress could provide additional targets for therapeutics in the treatment of AKI. 

A major obstacle in chemotherpy is the development of drug chemoresistance, leading to therapeutic failure. The mechanisms underlying Cp chemoresistance are multifactorial. E_2_ contributes to the development of Cp chemotherapy resistance by inhibiting Cp-induced apoptosis in lung cancer cells via a nonclassical ER signaling pathway [68]. Recent evidence indicates that GPER1 contributes to the sensitivity of certain cell types to Cp treatment. Specifically, it has been recently shown that GPER1 knockdown or antagonism enhances the sensitivity of gastric cancer cells to Cp, suggesting that GPER1 can be a target to improve Cp chemotherapeutic efficacy against gastric tumors developing chemoresistance [69]. Similarly, it has been shown that exposure to the estrogen disruptor, 2,2′,4,4′-tetrabromo diphenyl ether-47 (BDE-47), attenuates the sensitivity of endometrial carcinoma cells to Cp, thus eliciting chemoresistance, at least in part via a GPER1 signaling pathway [70]. Overall, the contribution of sex hormonal receptors in the development of Cp chemoresistance is poorly understood. Additional studies targeting the unraveling of the role of classical and nonclassical ER signaling in mediating therapy resistance to Cp can improve therapeutic outcomes in cancer patients. 

In addition to mediating estrogenic actions, it has been demonstrated that GPER1 mediates aldosterone downstream signaling actions in vascular and renal tissues [71,72,73]. Whether aldosterone binds directly to GPER1 or initiates a crosstalk between GPER1 and mineralocorticoid receptors is questionable [15,74]. In addition, aldosterone induces its biosynthesis via GPER1 activation in adrenocortical cells, which contributes to blood pressure regulation [75,76]. However, the contribution of aldosterone to GPER1-HO-1 interaction in Cp-induced nephropathy remains to be determined.

Excessive generation of ROS has been implicated in AKI pathophysiology [77]. Cp-induced renal injury has been associated with mitochondrial abnormalities, excessive endoplasmic reticulum stress, and increased ROS generation in renal proximal tubules [78,79,80]. We show that Cp treatment to WT mice evokes an increase in renal apoptosis that did not reach statistical significance. Further, the current study demonstrates that GPER1 deletion increases TUNEL-positive nuclei in the kidneys from saline and Cp-treated mice, highlighting a crucial role for GPER1 in suppressing apoptotic machineries. This finding is consistent with former studies reporting antiapoptotic actions for GPER1 [34,66,81]. GPER1 activation by G1 reduces plasma ROS levels in a rat model of doxorubicin cardiotoxicity [66] and increases the level of superoxide dismutase in H9C2 myocardial cells following ischemia reperfusion [81]. In addition, G1 inhibits the expression of endoplasmic reticulum stress-related proteins and apoptosis in the hippocampus following cerebral ischemia reperfusion [34]. In contrast, other studies had reported a permissive role for GPER1 in mediating oxidative stress. A role for GPER1 in mediating ROS and endothelial aging in renal arteries has been demonstrated [82]. In addition, Huang et al. showed that GPER1 mediates bisphenol A-induced ROS production in the human granulosa cell [83]. Thus, further studies are required to clarify the interaction between GPER1, mitochondrial bioenergetics, and oxidative stress as it relates to renal injury. Identifying GPER1 as an upstream regulator for ROS synthesis in the kidney may lead to the development of new therapies that can ameliorate oxidative damage in early stages of AKI. 

## 4. Materials and Methods

**Animal Studies.** The current studies utilized male mice, homozygous knockout for the GPER1 gene (homo) and heterozygous knockout for the GPER1 gene (het), and wild-type (WT) littermates obtained from our in-house colony. Knockout of GPER was confirmed by PCR genotyping of tail DNA. At the age of 14–16 weeks, mice were subjected to a single intraperitoneal injection of Cp (cisplatin, IP, 20 mg/kg body weight; pharmaceutical grade; NDC: 68001-283-27 from BluePoint Laboratories, Intas Pharmaceuticals, India) or normal saline (0.9% Sodium Chloride, pharmaceutical grade, NDC: 0409-4888-02 from Hospira, Lake Forest, IL, USA) as a vehicle (Figure 7). Cp and saline injections were administered between 9 am and 11 am. Mice were housed under a 12:12 h light–dark schedule with lights on at 6 am. Mice were maintained on a normal mouse chow (NIH-31 specification NSN 8710-01-005-8438 from Envigo, Indianapolis, IN, USA) and water was provided ad libitum.

Body weight was measured at the time of Cp/saline injection. Then, mice were weighed at the same time daily for 3 consecutive days (Figure 7). Three days after Cp/saline injections, the mice were euthanized between 9 am and 11 am. Euthanization was performed by an intraperitoneal injection of 2,2,2-tribromoethanol (1.25%, ip, T48402, Sigma-Aldrich, St. Louis, MO, USA). Kidneys and serum were processed for further analyses (Figure 1). Renal cortical tissues were preserved by snap freezing in liquid nitrogen for further analyses by Western blotting. 

**Measurement of serum biomarkers of renal injury.** SCr levels were determined using LC-MS/MS conducted at the Bioanalytical Core at the UAB’s O’Brien Core Center for AKI Research. Serum urea and NGAL levels were determined using the Bioassay Systems Urea Assay Kit (50-107-8333 from Fisher Scientific, Waltham, MA, USA) and NGAL ELISA kit (ab119602 from Abcam, Waltham, MA, USA), respectively, following the manufacturer’s instructions.

**Western blotting.** Western blotting was performed as previously described [7]. Briefly, kidneys were homogenized in radioimmunoprecipitation assay lysis buffer (RIPA) containing protease and phosphatase inhibitor cocktail (78441 from ThermoFisher Scientific, Waltham, MA, 02451 USA). Protein concentration was measured using the Bicinchoninic Acid (BCA) assay kit (23227 from Thermo Scientific, Waltham, MA, USA). Protein samples (50 µg) were resolved on 10% gels and transferred through Genie Electrophoretic Transfer (Idea Scientific Company, Minneapolis, MN, USA) to nitrocellulose membranes (1215483 from GVS Life Sciences, Panorama City, CA, USA). Blots were blocked in TBS Tween-20 containing 10% nonfat dry milk (1706404 from BioRad Laboratories, Hercules, CA, USA). The blots were incubated overnight in 5% milk containing the following primary antibodies: goat anti-KIM-1 (AF1817 from R&D systems, Minneapolis, MN, USA; lot- KCA0319121; 0.25 μg/mL), goat anti-NGAL (AF1857 from R&D systems, Minneapolis, MN, USA; lot-JZP0418071; 1:1000), and rabbit anti-HO-1 (ab68477 from Abcam, Waltham, MA, USA; lot-GR260152–1; 1:1000). For secondary antibody incubations, Alexa Fluor 790 donkey anti-rabbit IgG (711-655-152 from Jackson ImmunoResearch Laboratories, West Grove, PA, USA) and Alexa Fluor 680 donkey anti-goat IgG (705-625-147 from Jackson ImmunoResearch Laboratories, West Grove, PA, USA) were used. The blots were imaged using Licor’s Odyssey Fc infrared Imaging System and Image Studio Lite Ver5.2 software (Version: 5.2; Licor Biotechnology (https://www.licor.com/corp/history, accessed on 6 February 2021); Lincoln, NE, USA). Following transfer, the blots were stained with Ponceau S solution (P7170 from Sigma-Aldrich, St. Louis, MO, USA) for confirmation of equal loading. Densitometry on the bands of interest was carried out using ImageJ software (https://imagej.nih.gov/ij/; Available online: https://imagej.nih.gov/ij/download.html; accessed on 6 February 2021) and normalized to the Ponceau S staining. Densitometry values are represented as arbitrary units. Because the number of samples precluded the inclusion of all samples on a single blot, anchor protein was added across all gels for gel-to-gel comparison.

**Measurement of serum cytokine levels.** Serum cytokines were determined using the V-PLEX Proinflammatory Panel 1 Mouse Kit (K15048D-1 from Meso Scale Discovery, Rockville, MD, USA) following the manufacturer’s recommendations. 

**Histological assessment of renal injury.** Kidney tissues were fixed in 4% paraformaldehyde overnight at 4 °C and processed for paraffin embedding by the UAB Pathology Core Laboratory. Tissues were cut longitudinally into 5 μm thick sections and mounted on Superfrost slides. Kidney structures were stained with the periodic acid-Schiff (PAS) stain (22-110-645 from ThermoFisher Scientific, Waltham, MA, USA) using the manufacturer’s instructions. Images (~10/animal) of each kidney were obtained on a Keyence BZ-X710 microscope (Keyence Corporation of America, Itasca, IL, USA) with BZ-X software BZ-H3AE at a magnification of 20×. Images were taken at random across one section for each animal. Individual images were then scored in a blinded fashion using a scale of 0–4 (0 = no injury, 1 = 1/4 of the field-contained damage, 2 = 1/2 of the field-contained damage, 3 = 3/4 of the field-contained damage, 4 = the entire field-contained damage). Scores obtained from all images of each animal were averaged to provide an animal score. The mean score of each group was then calculated from the animal scores and reported as mean ± SEM. 

**Terminal deoxynucleotidyl transferase-mediated dUTP nick-end labeling (TUNEL) assay.** Detection of apoptotic cells was performed using the TACS2 TdT-DAB in situ Apoptosis Detection Kit (4810–30-K from R&D Systems) following the manufacturer’s instructions. TUNEL-positive nuclei were quantified in 5 μm thick paraffin-embedded kidney sections in a blinded fashion encompassing 8–9 microscopic fields of the renal cortex and outer medulla per animal. TUNEL-positive counts are reported as an average of the count of 8–9 random microscopic fields per kidney. Experimental group means were reported as mean ± SEM.

**Immunofluorescence staining.** Assessment of cell proliferation was performed by Ki67 immunofluorescence staining. Kidney tissues were fixed in 4% paraformaldehyde overnight at 4 °C, cryopreserved in 20% sucrose solution (BP220-1 from Fisher Scientific, Waltham, MA, USA) at 4 °C overnight, and embedded in Tissue-Plus O.C.T. compound (4585 from Fisher Scientific, Waltham, MA, USA). Sections 5 μm thick were immunostained as previously described [7]. Sections were incubated overnight with rabbit anti-Ki67 antibody (ab15580 from Abcam, Waltham, MA, USA; 1:200) and, subsequently, with secondary antibody AlexaFluor 594 donkey anti-rabbit (711-585-152 from Jackson Immuno Research, West Grove, PA, USA; 1:200). Slides were stored at −20 °C until imaged. Images were acquired by using a Keyence BZ-X710 microscope (Keyence Corporation of America, Itasca, IL, USA) with BZ-X800 software. Ki67-positive nuclei were counted blindly in 4–5 random fields in each kidney section. Experimental group means were reported as mean ± SEM.

**Statistics.** GraphPad Prism version 9 (GraphPad Software Inc., San Diego, CA, USA) was used for statistical analysis. The data values presented indicate means ± SEM. Statistical comparisons were conducted using Student’s *t*-test, two-way ANOVA, or repeated-measures two-way ANOVA with Sidak’s post hoc test for multiple comparisons. Statistical tests used for each dataset are specified in the associated figure legend. *P* values for each dataset are specified within the associated figure. *p* values of less than 0.05 were deemed statistically significant. 

## 5. Conclusions

In summary, we provide evidence that the global GPER1 deletion exacerbates renal apoptosis and attenuates Cp-induced upregulation of renal HO-1, independent of the extent of acute renal injury evoked by Cp treatment. GPER1 knockout did not exacerbate the Cp-induced AKI, as demonstrated by serological, histological, and renal tissue markers of injury, findings that are in agreement with previous reports in an acute kidney injury model [41]. However, our data do not support a role for GPER1 in the renal injury induced by Cp in male mice, and future efforts are needed to test whether the female sex, aging, different times of dosing for Cp, or tissue harvesting uncover a role for GPER1 in Cp-induced nephrotoxicity. Our findings suggest that GPER1 may play an anti-apoptotic role and a permissive role for HO-1 cytoprotective actions in AKI. GPER1-induced regulation of HO-1 after Cp treatment may offer a novel therapeutic target for the treatment of AKI. Future studies are needed to identify the significance of GPER1 regulation of HO-1 induction in mediating AKI and determining the underlying mechanisms of these GPER1-HO-1-interactions in a sex-specific manner.

## Figures and Tables

**Figure 1 ijms-23-08284-f001:**
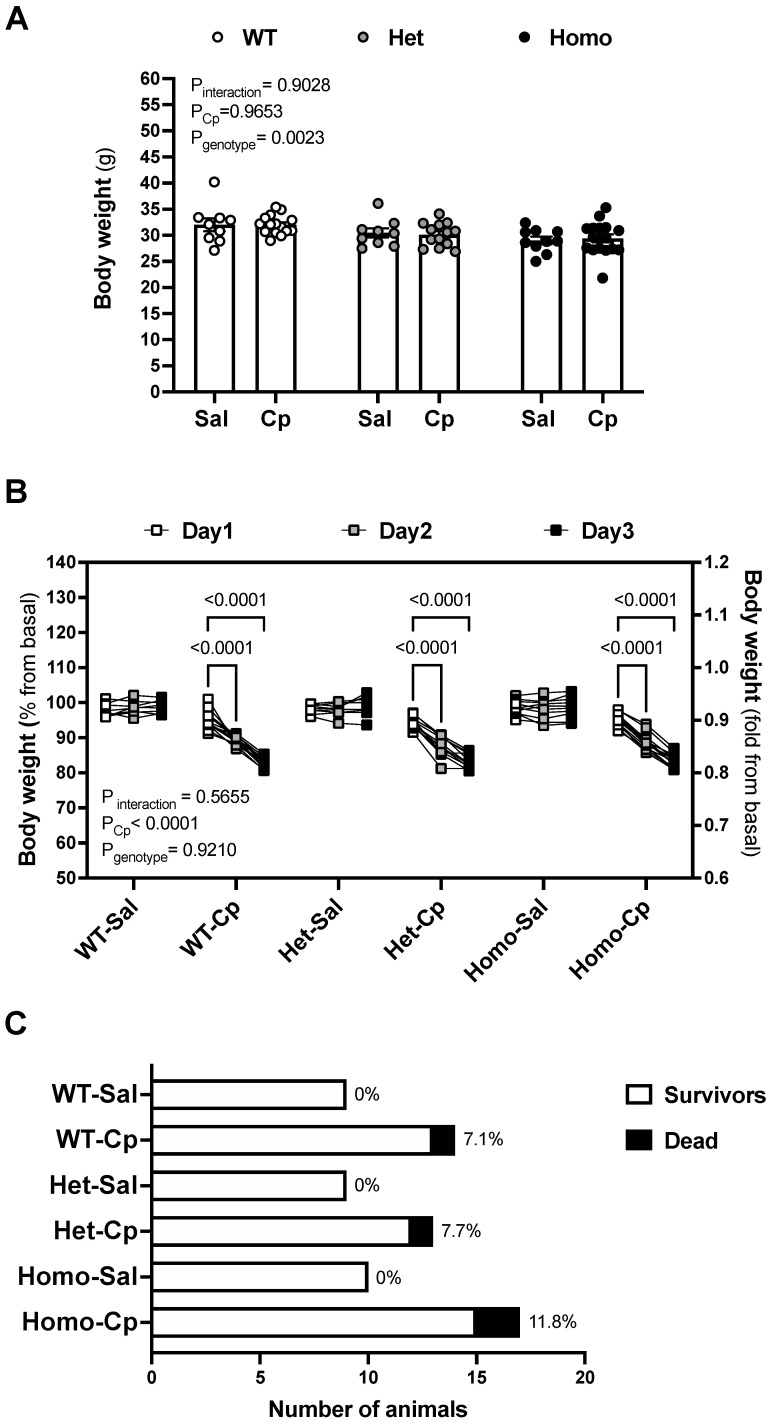
**Cisplatin-induced body weight loss and mortality rate.** Basal body weight on day 0 (**A**) and percentage change in body weight, relative to basal (day 0) values, on days 1–3 after cisplatin (Cp) or saline injections (**B**) were measured in male mice, homozygous (homo) and heterozygous (het) knockout for GPER1 gene, and wild-type (WT) littermates. The number of animals (WT, het, and homo) that survived or died in response to Cp or saline injections (**C**) was recorded during the first 3 days after the injections. Mice were injected with Cp (20 mg/kg) or saline on day 0. Data are presented as mean ± SEM (n = 9–17 mice/group). Statistical comparisons were performed by two-way ANOVA (**A**) or repeated measures two-way ANOVA (**B**) with Sidak’s post hoc test for multiple comparisons. *p* values for ANOVA and post hoc test results are displayed on the figures. For the three-way ANOVA, only *p* < 0.05 is displayed for clarity. Cp, cisplatin; Het: heterozygous knockout for GPER1 gene; Homo, homozygous knockout for GPER1 gene; Sal, saline; WT, wild-type.

**Figure 2 ijms-23-08284-f002:**
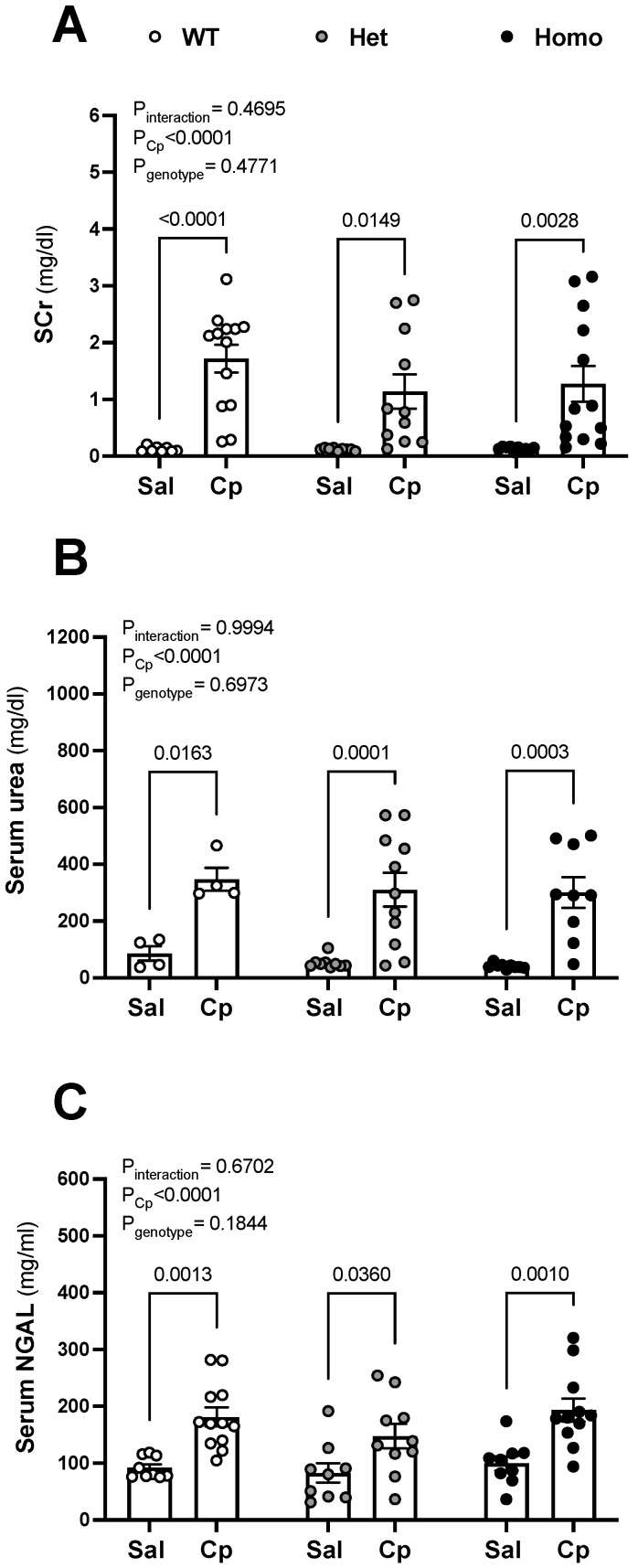
**Cisplatin-induced rise in serum levels of kidney injury markers.** Serum levels of creatinine (SCr, (**A**)), urea (**B**), and neutrophil gelatinase-associated lipocalin (NGAL, (**C**)) on day 3 after cisplatin (Cp, 20 mg/kg) or saline injections were measured in male, mice homozygous (homo) and heterozygous (het) knockout for GPER1 gene, and wild-type (WT) littermates. Data are presented as mean ± SEM (n = 4–13 mice/group). Statistical comparisons were performed by two-way ANOVA with Sidak’s post hoc test for multiple comparisons (**A**–**C**). *p* values for ANOVA and post hoc test results are displayed on the figures. Cp, cisplatin; Het: heterozygous knockout for GPER1 gene; Homo, homozygous knockout for GPER1 gene; Sal, saline; SCr, serum creatinine; WT, wild-type.

**Figure 3 ijms-23-08284-f003:**
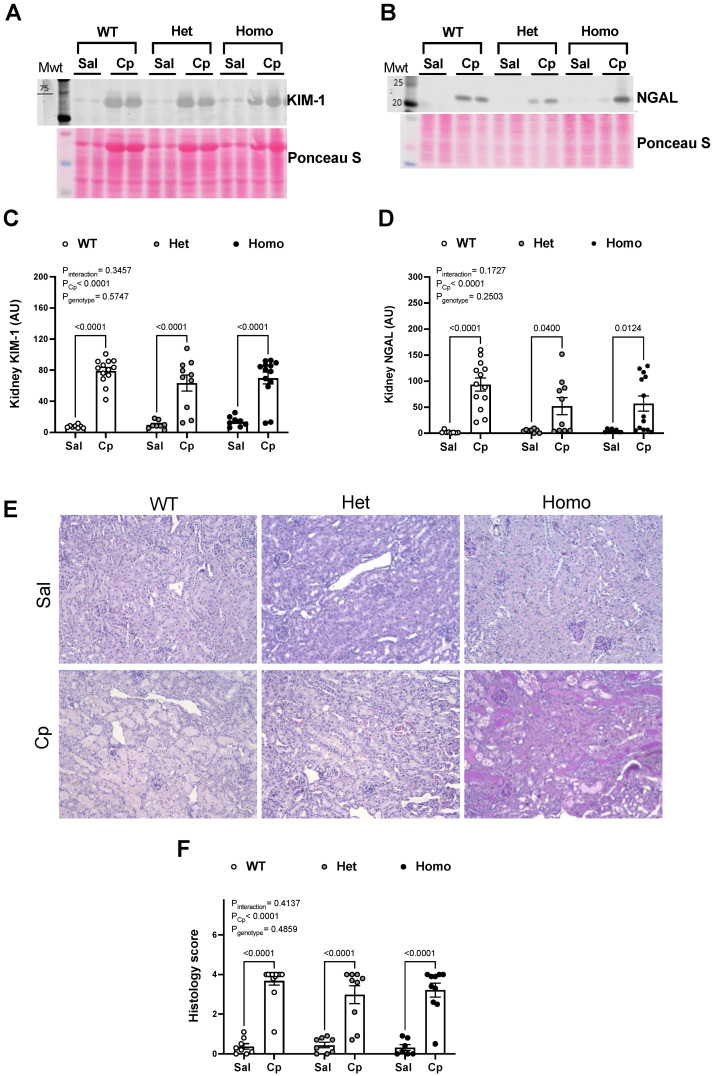
**Cisplatin-induced rise in kidney levels and histological indexes of renal tubular injury.** Representative images for Western blots on total kidney lysates for kidney injury molecule 1 (KIM-1, (**A**)) and neutrophil gelatinase-associated lipocalin (NGAL, (**B**)) measured on day 3 after cisplatin (Cp, 20 mg/kg) or saline injections in male mice, homozygous (homo) and heterozygous (het) knockout for GPER1 gene, and wild-type (WT) littermates. Densitometric values (AU, arbitrary units) for kidney levels of KIM-1 (**C**) and NGAL (**D**) were normalized to the Ponceau S staining. Representative histological images of periodic acid-Schiff (PAS)-stained kidney sections from male homo, het, and WT mice on day 3 after Cp or saline injections are shown (**E**). Scale bar = 200 µm. Quantification using a semiquantitative scale (0–4) of indexes of renal injury is demonstrated (**F**). Data are presented as mean ± SEM (n = 7–13 mice/group). Statistical comparisons were performed by two-way ANOVA with Sidak’s post hoc test for multiple comparisons (**C**,**D**,**F**). *p* values for ANOVA and post hoc test results are displayed on the figures. AU, arbitrary units; Cp, cisplatin; Het: heterozygous knockout for GPER1 gene; Homo, homozygous knockout for GPER1 gene; KIM-1, kidney injury molecule 1; NGAL, neutrophil gelatinase-associated lipocalin; PAS, periodic acid Schiff; Sal, saline; WT, wild-type.

**Figure 4 ijms-23-08284-f004:**
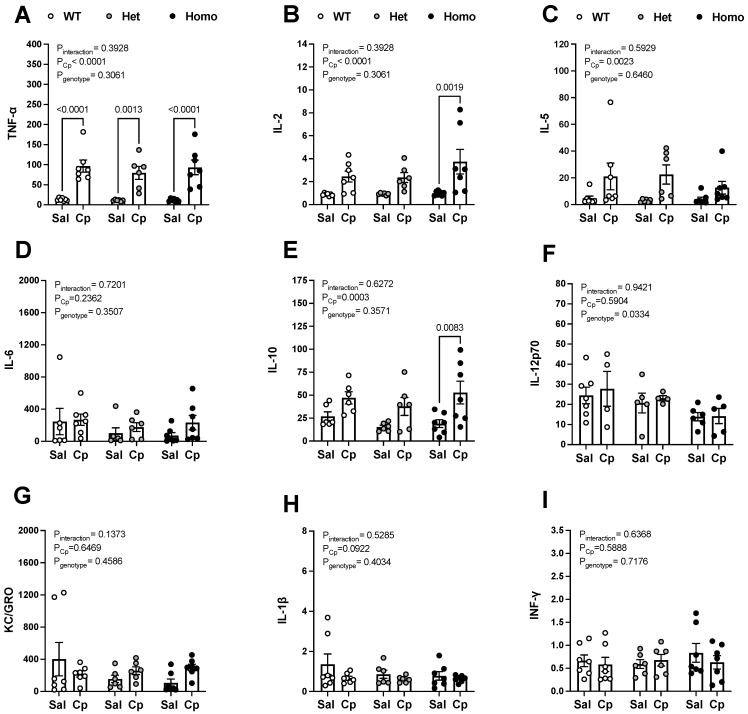
**Effect of GPER1 deletion on serum levels of inflammatory cytokines in cisplatin-treated mice.** Serum levels of tumor necrosis factor-α (TNF-α, (**A**)), interleukin-2 (IL-2, (**B**)), IL-5 (**C**), IL-6 (**D**), IL-10 (**E**), IL-12p70 (**F**), keratinocyte chemoattractant-growth-regulated oncogene (KC/GRO, panel (**G**)), IL-1β (**H**), and interferon-γ (INF-γ, (**I**)) measured in pg/mL on day 3 after cisplatin (Cp, 20 mg/kg) or saline injections in male mice, homozygous (homo) and heterozygous (het) knockout for GPER1 gene, and wild-type (WT) littermates are demonstrated. Data are presented as mean ± SEM (n = 4–7 mice/group). Statistical comparisons were performed by two-way ANOVA with Sidak’s post hoc test for multiple comparisons (**A**–**I**). *p* values for ANOVA and post hoc test results are displayed on the figures. Cp, cisplatin; Het: heterozygous knockout for GPER1 gene; HO-1, heme oxygenase; Homo, homozygous knockout for GPER1 gene; INF-γ, interferon-γ; IL, interleukin; KC/GRO, keratinocyte chemoattractant-growth-regulated oncogene; Sal, saline; TNF-α, tumor necrosis factor-α; WT, wild-type.

**Figure 5 ijms-23-08284-f005:**
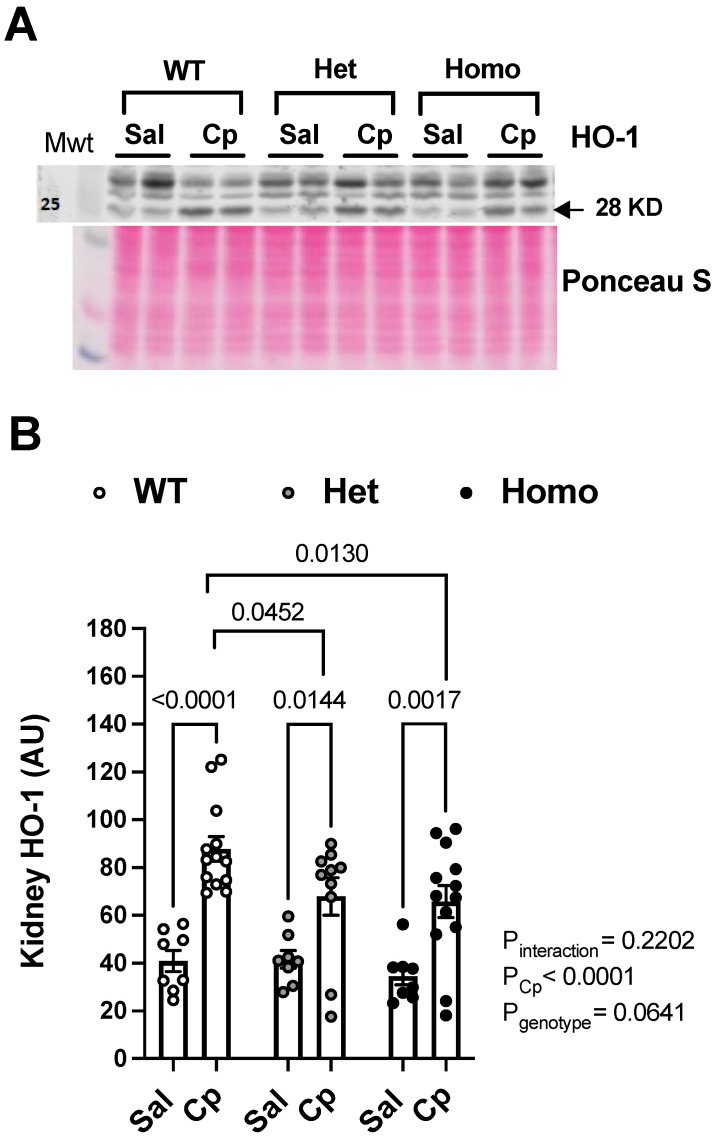
**GPER1 deletion blunts cisplatin-induced rise in kidney heme oxygenase-1.** Representative images for Western blots on total kidney lysates for heme oxygenase-1 (HO-1, (**A**)) measured on day 3 after cisplatin (Cp, 20 mg/kg) or saline injections in male mice, homozygous (homo) and heterozygous (het) knockout for GPER1 gene, and wild-type (WT) littermates. Densitometric values (AU, arbitrary units) for kidney levels of HO-1 (**B**) were normalized to the Ponceau S staining. Data are presented as mean ± SEM (n = 8–13 mice/group). Statistical comparisons were performed by two-way ANOVA with Sidak’s post hoc test for multiple comparisons (**B**). *p* values for ANOVA and post hoc test results are displayed on the figures. AU, arbitrary units; Cp, cisplatin; Het: heterozygous knockout for GPER1 gene; HO-1, heme oxygenase; Homo, homozygous knockout for GPER1 gene; Sal, saline; WT, wild-type.

**Figure 6 ijms-23-08284-f006:**
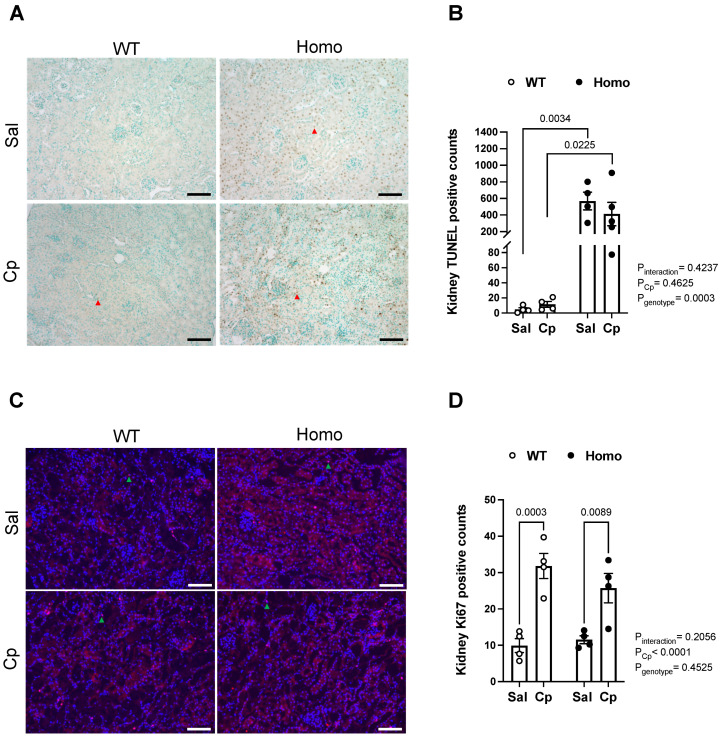
**Effect of GPER1 deletion on cell death and proliferation in cisplatin-treated mice.** Representative images for TUNEL assay performed for kidney sections obtained from cisplatin (Cp, 20 mg/kg) or saline-treated male mice, homozygous (homo) knockout for GPER1 gene, and wild-type (WT) littermates on day 3 of the treatment are demonstrated (**A**). Red arrowheads point to TUNEL-positive nuclei. Quantification of TUNEL-positive nuclei in different experimental groups is demonstrated (**B**). Representative images for Ki67 immunostaining performed for kidney sections obtained from cisplatin (Cp, 20 mg/kg) or saline-treated homo and WT male mice on day 3 of the treatment are demonstrated (**C**). Green arrowheads point to TUNEL-positive nuclei. Scale bar represents 200 µm. Quantification of Ki67-positive nuclei in different experimental groups is demonstrated (**D**). Data are presented as mean ± SEM (n = 4–5 mice/group). Statistical comparisons were performed by two-way ANOVA with Sidak’s post hoc test for multiple comparisons. *p* values for ANOVA and post hoc test results are displayed on the figure. Cp, cisplatin; Homo, homozygous knockout for GPER1 gene; Sal, saline; TUNEL, Terminal deoxynucleotidyl transferase-mediated dUTP nick-end labeling; WT, wild-type.

**Figure 7 ijms-23-08284-f007:**
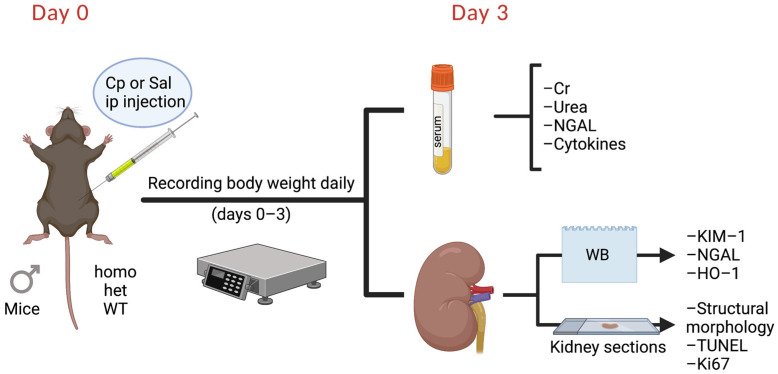
Experimental timeline. Cp, cisplatin; Cr, creatinine; het: heterozygous knockout for GPER1 gene; HO-1, heme oxygenase-1; homo, homozygous knockout for GPER1 gene; KIM-1, kidney injury molecule 1; NGAL, neutrophil gelatinase-associated lipocalin; Sal, saline; TUNEL, Terminal deoxynucleotidyl transferase-mediated dUTP nick-end labeling; WB, Western blotting; WT, wild-type. Created with BioRender.com.

## Data Availability

All data generated or analyzed during this study are included in this published article.

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
