# Peer review of "Does G Protein-Coupled Estrogen Receptor 1 Contribute to Cisplatin-Induced Acute Kidney Injury in Male Mice?"

_ijms, 2022, doi:10.3390/ijms23158284_

Round 1

Reviewer 1 Report

There are a lot of factors that cause the kidney injury including the physical construction or the chemical therapy mediated damage .Here the authors state a frequent antitumor compound, cisplatin, which could lead to the acute kidney injury, but the mechanism is unclear. They focused on a gene named GPER1,which could play some role in this process. GPER1 deletion did not change the injury markers expression but the GPER1 plays the anti-apoptotic role and a permissive role for HO-1 cytoprotective actions in AKI. GPER1-induced regulation of HO-1 after Cp treatment may offer a novel therapeutic target for the treatment of AKI.

Major concerns:

The authors could not get enough evidence to reach the conclusion that GPER1 contribute to cisplatin-induced acute kidney injury. From figure 2-figure 5 the authors only discovered the cisplatin could induce the kidney injury by different methods. At last, the authors provide limited results that GPER1 could regulate or influence the HO-1 expression. More results needed to confirm how the GPER1 regulates the HO-1,and the relationship between the HO-1 and apoptosis/injury etc.

Minor concerns:

In figure 3/4/5, you compared the expression of SCR/Serum urea/NGAL between the sal and cp, but no comparison in the WT/HET/Homo. If there is no significant difference, p value also could be labeled.

Author Response

We would like to thank the reviewers and editors for their helpful comments. We have made changes accordingly that have clarified and strengthened the manuscript. Responses to reviewers’ comments and changes to the text of the revised manuscript have been made in red.

Reviewer 1.

Major concerns:

The authors could not get enough evidence to reach the conclusion that GPER1 contribute to cisplatin-induced acute kidney injury. From figure 2-figure 5 the authors only discovered the cisplatin could induce the kidney injury by different methods. At last, the authors provide limited results that GPER1 could regulate or influence the HO-1 expression. More results needed to confirm how the GPER1 regulates the HO-1, and the relationship between the HO-1 and apoptosis/injury etc.

Response. The relationship between HO-1 and GPER1 is likely to be multifaceted and complex. HO-1 has anti-apoptotic, anti-inflammatory and antioxidant effects in AKI. These effects are conferred both by the byproducts of the enzymatic reaction as well as by the cleaved form of HO-1. HO is a stress response enzyme that detoxifies heme moieties of heme containing proteins. This enzymatically active form resides in the membrane of the endoplasmic reticulum. A cleaved form of HO-1 has more recently been identified that is not thought to contain the enzymatic activity, but rather confers transcriptional effects that leads to expression of downstream mediators of anti-apoptotic, anti-inflammatory and antioxidant actions. HO-1 expression is substantially upregulated in the proximal tubule of the kidney, the primary site of apoptosis, after cisplatin-induced AKI. Knockout of HO-1 increases the presence of apoptotic cells in the kidney after a model of AKI (PMID: 10807584). However, HO-1 has also been shown to be expressed by macrophages and other monocytes, as well as in bone marrow stem cells, each of which has been shown to play a role in modifying the manifestation of AKI. Changes in the complement of resident and infiltrating macrophages in the kidney after AKI is also altered by changes in the expression of HO-1 (PMID: 25677389). Mesenchymal stem cells have been shown to be protective in AKI, but lose that protection in the absence of HO-1 (PMID: 21048024). These studies have only been conducted in males, but sex differences in the expression of HO-1 cleaved form are suggestive of an estrogen effect (PMID: 28679590). Estrogen effects on HO-1 in have been noted in the heart (PMID: 22730333). The complexity of the actions of HO-1 coupled with the unknown roles of GPER1 require further study to target each of the facets of HO-1 activity in AKI, but these studies are beyond the scope of this study. Furthermore, given the short time for revision, we were not able to perform more experiments.

Minor concerns:

In figure 3/4/5, you compared the expression of SCR/Serum urea/NGAL between the sal and cp, but no comparison in the WT/HET/Homo. If there is no significant difference, p value also could be labeled.

Response. Thanks for your comment. We have provided p values for post-hoc analysis conducted between WT and cisplatin-treated groups, since ANOVA results showed a significant effect for treatment (fig 3-5). For these figures, two-way ANOVA results are displayed on the upper left side of each panel. Since ANOVA results did not show significant interaction or significant effect for genotype, we did not perform post-hoc test analysis between WT, Het and Homo groups (fig 3, 4, 5A-E, 5G-I). For fig 5F, we now clarify in the paper that while a significant genotypic effect was noted by ANOVA results for fig 5F, post-hoc test between WT/Het/Homo did not reach statistical significance. We also added p values for the post-hoc test analysis (lines 259-261).

Reviewer 2 Report

The manuscript entitled "Does G protein-coupled estrogen receptor 1 contribute to cisplatin-induced acute kidney injury in male mice" by Gohar et al suggest that GPER1 plays cytoprotective and anti-apoptotic roles in AKI. The manuscript in general is well written and the study design is well described. However, there are some language needs to be improved. The major concerns are:

1) The overall observation from the presented data suggests that GPER 1 probably does not play any role in cisplatin-induced AKI. However, this is important so as to guide others not pursue this pathway for future mechanistic studies. This should be clearly indicated in the conclusions.

2) The authors cited several studies that provided the role for GPER 1 in other diseases (page 2, lines 54-57). These studies are interesting. However, the authors should include some information whether these roles are solely estrogen-dependent or that GPER1 can be activated by steroid hormones other than estrogen.

3) GPER1 has been shown to be involved in several neuronal functions. Did the animals showed any neurological and/or behavioral deficiencies? How long did the animals survive?

4) In figure 6, the western blot shows that cisplatin decreased HO-1 expression in the WT mice compared to the saline treated WT mice while in the hetero- and homozygotes HO-1 expression was higher in cisplatin-treated mice compared to the saline mice. However, the bar diagram shows increased HO-1 expression in cisplatin-treated WT mice. The description in results match the bar diagram data. The authors need to clarify these data.

5) In figure 7C, it appears that the homozygous mice already has higher Ki67 staining (if the red stain is for Ki67) and that there is no further increase in Ki67 staining. Am I reading this figure correctly? The authors need to clarify these data.

6) Overall the discussion must be modified to make it clear that at least in the male mice, GPER1 plays no role in cisplatin-induced AKI.

Author Response

We would like to thank the reviewers and editors for their helpful comments. We have made changes accordingly that have clarified and strengthened the manuscript. Responses to reviewers’ comments and changes to the text of the revised manuscript have been made in red.

Reviewer 2.

1)The overall observation from the presented data suggests that GPER1 probably does not play any role in cisplatin-induced AKI. However, this is important so as to guide others not pursue this pathway for future mechanistic studies. This should be clearly indicated in the conclusions.

Response. We agree with the reviewer that our data does not support a role for GPER1 in cisplatin-induced and we have revised the manuscript to state that clearly (lines 26, 349, 350). We also highlighted that the lack of evidence for GPER1 contribution to the extent of cisplatin-induced renal injury is limited to the experimental design used in the present study. Whether the female sex, aging, different times of dosing for cisplatin or tissue harvesting uncovers a role for GPER1 in cisplatin-induced nephrotoxicity remain to be tested (lines 490-493).

2)The authors cited several studies that provided the role for GPER1 in other diseases (page 2, lines 54-57). These studies are interesting. However, the authors should include some information whether these roles are solely estrogen-dependent or that GPER1 can be activated by steroid hormones other than estrogen.

Response. We thank the reviewer for bringing up this important point. Evidence points to a contribution for GPER1 in mediating aldosterone downstream signaling. As suggested, we have now elaborated on the contribution of estrogen vs aldosterone in regulating GPER1 activity (lines 455-461).

3) GPER1 has been shown to be involved in several neuronal functions. Did the animals showed any neurological and/or behavioral deficiencies? How long did the animals survive?

Response. The reviewer is correct that GPER1 is widely expressed in the central and peripheral nervous system. Its activation significantly contributes to mitigating stress-induced anxiety (PMID: 23669322), improves memory (PMID: 29421611), induces neuroprotective effect on dopaminergic neurons (PMID: 24726471), protects hippocampal and striatal neurons from ischemic damage (PMCID: PMC3587667) and mediates anti-inflammatory effect in ischemic stroke (PMID: 27127723). However, in our study we did not see any obvious signs or symptoms of neurological and/or behavioral deficiencies in our mice (14-16 weeks old) as compared to the wild-type control mice. Additional ageing-related studies are required to determine whether GPER1 deletion impacts survival rate. Although we did not evaluate the contribution of renal innervation in the current study as it relates to cisplatin-induced renal injury, neurological issues in this animal model are not well established.

4) In figure 6, the western blot shows that cisplatin decreased HO-1 expression in the WT mice compared to the saline treated WT mice while in the hetero- and homozygotes HO-1 expression was higher in cisplatin-treated mice compared to the saline mice. However, the bar diagram shows increased HO-1 expression in cisplatin-treated WT mice. The description in results match the bar diagram data. The authors need to clarify these data.

Response. We apologize for the confusion. We have an arrow pointing to the 28-KD band on the Western blot, which is the HO-1 band that was detected and analyzed. The 28-KD band for HO-1 has been previously shown to be modulated in cisplatin-induced nephrotoxicity (PMCID: PMC5625098). We have highlighted that now in the manuscript (lines 281-282). Similar to the data presented on the bar diagram data, the analyzed HO-1 band demonstrates increased expression of HO-1 in response to cisplatin treatment in WT animals.

5) In figure 7C, it appears that the homozygous mice already has higher Ki67 staining (if the red stain is for Ki67) and that there is no further increase in Ki67 staining. Am I reading this figure correctly? The authors need to clarify these data.

Response. Thanks for your comment. Ki67 is a nuclear stain and cytoplasmic staining is disregarded and it has been shown that Ki67 nuclear expression is proportional to the mitotic count. While the representative image for Ki67 staining shows non-specific background staining, specific staining was only quantified by counting Ki67-postitive nuclei as indicated in figure legend (line 330-331) and under the Methods section (lines 165-166).

6) Overall the discussion must be modified to make it clear that at least in the male mice, GPER1 plays no role in cisplatin-induced AKI.

Response. Thanks for your comment. We now state clearly that our data does not support a role for GPER1 in cisplatin-induced AKI in male mice (lines 349, 350, 490-493).

Round 2

Reviewer 1 Report

The authors get the conclusion that GPER1 has no role in mediating Cp-induced renal injury and overturn the previous conclusions by the publications.

In the fig6, internal reference proteins such as the Gapdh or Actin should be used for reference.

For the KO mice, provide the genotyping identification and evidence that the GPER1 is knocked out by the western blot or other method.

In vitro study could be performed to verify the regulation of HO-1 by the GPER1 with the kidney cells.

Author Response

In the fig6, internal reference proteins such as the Gapdh or Actin should be used for reference.

Response. We appreciate the reviewer’s perspective regarding normalization of the western blotting data. In our experience, however, both GAPDH and actin can be changed by sex and/or by injury (as well as age) and thus are not useful as normalizing factors. For this reason, we use ponceau red staining of the membrane as it is a stable and reproducible normalizing factor. Because it stains all proteins, it provides a good surrogate of total protein. We typically do densitometry on the stained membrane excluding the ~60-70 kD range to avoid any interference by robust changes in albumin.

For the KO mice, provide the genotyping identification and evidence that the GPER1 is knocked out by the western blot or other method.

Response. We thank the reviewers for asking for this clarification. In maintaining the colony, we routinely use genotyping by PCR of the 555 and 730 bands to confirm the knockout. We have added a statement to clarify that in the Methods section of the paper (lines 78, 79). We have also uploaded supplemental data file for the reviewer’s reference providing genotyping results performed by PCR.

In vitro study could be performed to verify the regulation of HO-1 by the GPER1 with the kidney cells.

Response. We appreciate the suggestion of the reviewer. Unfortunately, given the 3-day turnaround for the paper revision, it is not feasible to conduct these studies for this manuscript. We intend to explore the impact of GPER1 knockout on HO-1 in vitro in a future study to better understand these dynamics. It should be noted that these studies will require examination of multiple inducers of injury to model AKI in vitro as well as use of inducers and inhibitors of HO-1 to explore the dynamics. These studies will necessarily have to be conducted on primary proximal tubule cells which will require generation of additional knockout animals. It is possible also that other cell types, different from the proximal tubule cells, may have contributed to the response. We anticipate that this future study will provide a more detailed understanding of the mechanism of interaction.

Reviewer 2 Report

The authors have sufficiently answered my comments. However, their response regarding Ki67 staining raises another question. The intracellular staining for Ki67 appears higher in CP-treated WT mice and in saline- or CP-treated homozygous mice than in the saline-treated WT mice. What is the significance of this staining pattern?  Can the authors explain this staining pattern?

Author Response

The authors have sufficiently answered my comments. However, their response regarding Ki67 staining raises another question. The intracellular staining for Ki67 appears higher in CP-treated WT mice and in saline- or CP-treated homozygous mice than in the saline-treated WT mice. What is the significance of this staining pattern?  Can the authors explain this staining pattern?

Response. We thank the reviewer for the careful review. It is not uncommon to see background staining in immunostaining experiments after cisplatin injury as there is significant cell death and cell derangement that occurs in response to the injury. Nonetheless, the specific staining of the nucleus exhibits clear morphology of an intact nucleus (both by position within the cell as well as the uniform size and shape) that allows for this measure to be reproducible.